# The Prognostic Significance of Puncture Timing to Survival of Arteriovenous Fistulas in Hemodialysis Patients: A Multicenter Retrospective Cohort Study

**DOI:** 10.3390/jcm8020247

**Published:** 2019-02-15

**Authors:** Su-Ju Lin, Chun-Wu Tung, Yung-Chien Hsu, Ya-Hsueh Shih, Yi-Ling Wu, Tse-Chih Chou, Shu-Chen Chang, Chun-Liang Lin

**Affiliations:** 1Department of Nephrology, Chang Gung Memorial Hospital, Chiayi 61363, Taiwan; 8902071@cgmh.org.tw (S.-J.L.); p122219@cgmh.org.tw (C.-W.T.); libra@cgmh.org.tw (Y.-C.H.); rita1608@gmail.com (Y.-H.S.); 2Graduate Institute of Clinical Medical Sciences, Chang Gung University, Taoyuan 33302, Taiwan; 3Kidney and Diabetic Complications Research Team (KDCRT), Chang Gung Memorial Hospital, Chiayi 61363, Taiwan; 4Research Services Center for Health Information, Chang Gung University, Taoyuan 33302, Taiwan; wyling83@hotmail.com; 5Clinical Informatics and Medical Statistics Research Center, Chang Gung University, Taoyuan 33302, Taiwan; sadricky@gmail.com; 6College of Medicine, Chang Gung University, Taoyuan 33302, Taiwan; 7Kidney Research Center, Chang Gung Memorial Hospital, Taipei 10507, Taiwan; 8Center for Shockwave Medicine and Tissue Engineering, Kaohsiung Chang Gung Memorial Hospital and Chang Gung University College of Medicine, Kaohsiung 83301, Taiwan

**Keywords:** arteriovenous fistula, hemodialysis, survival, maturation time, puncture time

## Abstract

(1) Background: A functional shunt is critical to hemodialysis, but the ideal timing of shunt cannulation is still not established. In this study, we assessed the association between ideal puncture timing and shunt survival. (2) Methods: This retrospective cohort study using data from the Taiwan Health and Welfare database, which included 26885 hemodialysis patients with arteriovenous fistulas from 1 July 2008 to 30 June 2012. Fistulas were categorized by functional maturation time, defined as the time from the date of shunt construction to the first successful cannulation. Functional cumulative survival, measured as the duration from the first puncture to shunt abandonment, was mainly regarded. (3) Results: The fistulas created between 91 and 180 days prior to the first cannulation had significantly greater cumulative functional survival (HR 0.883; 95% CI 0.792–0.984), and there was no more benefit on their survival from waiting more than 180 days (HR 0.957; 95% CI 0.853–1.073) for shunt maturity. (4) Conclusions: Our results showed that to achieve better long-term shunt survivals, fistulas should be constructed at least 90 days before starting hemodialysis. Notably, there was no additional benefit on waiting more than 180 days prior to cannulation.

## 1. Introduction

For patients with end-stage renal disease (ESRD), hemodialysis is the most prevalent renal replacement therapy. Having optimal vascular access with sufficient diameter and flow rate is essential for adequate hemodialysis. Arteriovenous fistula is the ideal one rather than other vascular access, such as arteriovenous graft, bridging long-term tunneled or short-term non-tunneled catheter, which increases the risk of thrombosis, central stenosis, infections, multiple hospitalizations, higher medical costs, and greater morbidity and mortality [1,2,3]. Consequently, maintaining functional fistulas is a serious challenge in the care of hemodialysis patients.

Researchers have attempted to identify variables associated with the maturation and survival of shunts, including anatomic locations and types of shunts, and demographic characteristics of patients, but the optimal puncture timing for shunts remains undetermined. One prospective observational study [4] analyzed incident dialysis patients in 309 facilities in France, Germany, Italy, Japan, Spain, the United Kingdom, and the US. The researchers found significant differences in clinical practices among countries, including the time from fistula creation to initial cannulation. The median time for this ranged from 25 to 98 days. In this study, reduced fistula survival was associated with cannulation within 14 days of fistula creation. Another population-based cohort study in Canada found that fistula creation at least 4 months before starting hemodialysis was associated with the lowest risk of sepsis and death; lower risk was credited to reduced use of hemodialysis catheters [5]. In contrast, Ng et al. [6] observed that fistulas received puncture after starting hemodialysis for more than 1 month had longer primary patency. Individualized programs for shunts were suggested due to diversity of maturation time in this study.

The current National Kidney Foundation Dialysis Outcomes Quality Initiative (KDOQI) [7] has endorsed the idea that the time required for fistula maturation varies among patients. The Work Group advised avoiding premature cannulating fistulas within the first month after construction; they also noted that a fistula should be placed for at least 6 months before hemodialysis begins, to allow shunt evaluation and additional time for revision if needed. [8,9] Nevertheless, the Japanese Society of Dialysis Therapy advised that fistula be constructed at least 2–4 weeks before dialysis, based on working group opinions [10]. In contrast, 2018 Clinical Practice Guidelines of the European Society for Vascular Surgery (ESVS) [11] recommended that arteriovenous fistulas should be considered for cannulation 4–6 weeks after creation. Among these practice guidelines, the optimal maturation time to puncture fistula and improve fistula survival is not established.

Although the importance of functional shunts on the morbidity and mortality of patients with ESRD is well known, only about 20% of patients start hemodialysis with mature shunt, even in developed countries. [12,13] One of the main reasons for this is that the appropriate time for shunt creation and cannulation is still undetermined. Therefore, we designed a study using the Taiwan Health and Welfare Database (National Health Insurance Research Database, NHIRD, previously) to estimate the optimal puncture time of a shunt to maintain its longevity.

## 2. Experimental Section

### 2.1. Source of Data

This retrospective cohort study was based on data from patients enrolled in the NHIRD for ESRD, which was provided from the Health and Welfare Data Science Center (HWDC). The National Health Insurance (NHI) system, a compulsory program in Taiwan, was initiated in 1995 and is co-funded by the Taiwanese government, employers, and beneficiaries. All residents, including foreigners, are mandated to join the NHI system if they have lived in Taiwan longer than 6 months. Thus, NHI data cover more than 99% of the population. The NHIRD consists of longitudinal medical records for all health beneficiaries in Taiwan. These records include patient characteristics, diagnoses, examinations, procedures, operations, and fees. The NHIRD databank was anonymized before being released for research use, to protect patient and physician privacy. The diagnoses in the NHIRD are coded using the format of the International Classification of Disease, Ninth Revision, Clinical Modification (ICD-9-CM), and claims data of all interventions and treatments also correspond to relevant codes. We obtained ethical approval for our study from the Chang Gung Medical Foundation Institutional Review Board, and the study was conducted in full compliance with national ethical guidelines. All data were anonymized by the Taiwan NHI system, thus the institutional review board determined that patient consent was not required.

### 2.2. Study Population

Using NHIRD data, we identified all patients with ESRD (ICD-9-CM code: 585) from 1 July 2008 to 30 June 2012. ESRD patients requiring maintenance long-term hemodialysis were confirmed by catastrophic illness certification profiles, which were applied by nephrologists; the status of ESRD had been documented by specialists in the NHI administration. We excluded patients younger than 20 years of age and patients who had undergone renal transplantation or peritoneal dialysis. The patients who have received arteriovenous fistulas creation were confirmed by relevant operative codes (AVF; procedure codes: 69032B, 69032C). In addition, patients were excluded who had unclear health records on shunt construction or cannulation or delayed shunt creation after initiating hemodialysis because we could not confirm the status of vascular access maturation. We followed patients until their deaths, deregistration, or the end of the study. The description of selection of study subjects is shown in Figure 1. A total of 26885 patients were enrolled and analyzed in this study. For the purpose of recognizing the most appropriate cannulation time of fistulas, we used quantile to group the subjects, and that of each group were within 16 days, between 17 and 31 days, between 32 to 97 days, and more than 98 days respectively. Further, to realize the association with fistula survivals and their waiting puncture time longer than 98 days, the subjects were regrouped by waiting puncture time as within 30 days, between 30 and 90 days, between 90 to 180 days, and more than 180 days on the basis of present guidance and clinical practicality.

### 2.3. Major Outcomes

As shown in Figure 2, these 26885 patients were categorized by functional maturation time, which was defined as the duration from the date of operation for fistula creation to the date of the first successful shunt cannulation. If the patients had more than once operative records of shunt, we identified the first one as our investigation to minimize mistakes about further revision or two-stage fistulas. For those patients who never received long-term tunneled catheter implantation prior to initiating hemodialysis, we selected the date of the first hemodialysis as the first successful shunt cannulation. For those who had long-term tunneled catheter implantation, the catheter was usually removed immediately after the first successful puncture in Taiwan, thus, we selected the date of permanent tunneled catheter removal to indicate the first date of successful cannulation of the shunt. Moreover, to eliminate the effects of technical failure during operation, we excluded reports of shunts with primary failures, defined as shunt abandonment before its first use for hemodialysis.

The primary outcome of this study was the functional cumulative survival of a shunt, defined as a shunt with functional maturation until abandonment or until reaching a censored event, including missing medical records in the NHIRD, patient death, or end of the study period. Functional primary patency, defined as the functional maturation of a shunt until any first intervention (endovascular or surgical) to maintain or restore blood flow, was also investigated and regarded as secondary outcomes.

### 2.4. Covariates

Baseline demographic and clinical data, including age, gender, comorbidities, and medications, were defined as covariates. The comorbidities, such as hypertension (HTN), diabetes mellitus (DM), myocardial infarction (MI), congestive heart failure (CHF), peripheral vascular disease (PVD), and cerebrovascular disease (CVD), were identified by the diagnostic codes in two or more consecutive outpatient records during the 6 months prior to shunt construction. For medications, including aspirin, clopidogrel (Plavix^®^), warfarin, and statins, we included the data when the drug had been used for at least 1 month, as indicated by outpatient records, before or after shunt creation.

### 2.5. Statistical Analysis

Data regarding basic demographic characteristics, comorbidities, and medications were presented as a number (percent) for categorical variables and mean ± standard deviation (SD) for continuous variables. Pearson’s chi-square test and analysis of variance were used to compare the differences in categorical and continuous variables, respectively. For each type of vascular accesses survival, Cox proportional hazard models and the Kaplan–Meier method were used to calculate hazard ratios (HRs) with 95% confidence intervals (CI). Analyses were conducted using SAS statistical software, version 9.4 (SAS Institute, Inc., Cary, NC, USA).

## 3. Results

### 3.1. Patient Characteristics

The demographic characteristics, comorbid diseases, and medication usage are listed in Appendix A and Table 1. As shown in Appendix A, the mean days in each quantile are 8.7 days, 23.7 days, 60.9 days and 256.1 days; the time spans are within 16 days, between 17 and 31 days, between 32 and 97 days and more than 98 days, respectively. Patients in the group quantile 1 were more likely to be comorbid with HTN, DM, MI, CHF, and PVD, and to be treated with aspirin. Patients in the group quantile 4 were more likely to have CVD, and to use clopidogrel and warfarin. Otherwise, the current KDOQI [7] suggested that a fistula should be placed for at least 6 months before hemodialysis begins. While grouping by using quantile, it was hard to evaluate the survival benefits of fistulas with waiting cannulation for longer than 98 days, between 98 and 6 months or longer. Consequently, based on present guidance and clinical practicality, the subjects were regrouped by waiting puncture time as within 30 days, between 30 and 90 days, between 90 to 180 days, and more than 180 days on the. Table 1 demonstrated that the functional maturation time of fistulas in most patients was within 30 days (49.6%). Patients with fistulas with functional maturation time within 30 days were more likely to be comorbid with HTN, DM, MI, CHF, and PVD, and to be treated with aspirin and statins. Patients with fistulas with functional maturation times greater than 180 days were more likely to have HTN, DM, CHF, PVD, CVD, and to use warfarin, but not MI, aspirin, or statins.

### 3.2. Functional Shunt Survivals

Figure 3 summarizes the patency of shunts studied. The 1-year, 3-year, and 5-year functional cumulative survival rates of the fistulas were 81.68%, 74.07%, and 69.54%. The 1-year, 3-year and 5-year functional primary patency rates were 54.20%, 36.51%, and 27.21%, respectively.

### 3.3. Optimal Functional Maturation Time of Shunts and Association between Covariates

The analysis of the primary outcome, from shunt cannulation to abandonment, is shown in Appendix A, Table 2 and Figure 3. Appendix A illustrates that the fistulas, which were punctured 98 days later after creation, had better cumulative functional survivals (HR 0.899; 95% CI 0.82–0.985; *p* = 0.0226). Furthermore, as shown in Table 2 and Figure 3, the fistulas with functional maturation time between 91 and 180 days had the greatest cumulative functional survival (HR 0.883; 95% CI 0.792–0.984; *p* = 0.0237), while the cumulative functional survival of fistulas with functional maturation time more than 180 days (HR 0.957; 95% CI 0.853–1.073; *p* = 0.4515) was not better. Patients who had fistulas with poorer functional cumulative survival were significantly more likely to be older, female, to have a history of DM, CHF, and to be using warfarin. This finding indicated that the patients with a fistula, which was created at least 3 months prior to hemodialysis, have less chance of shunt abandonment.

Appendix A, Table 3 and Figure 4 demonstrate the results of analysis of the fistulas from cannulation to the first intervention for maintaining adequate function. Appendix A displayed that the fistulas with functional maturation time within 16 days and between 17 and 31 days had poorer functional primary patency (HR 1.239; 95% CI 1.184–1.298; *p* < 0.0001 and HR 1.202; 95% CI 1.148–1.254; *p* < 0.0001). Similarly, as shown in Table 3, functional maturation times within 30 days had significantly the worst functional primary patency (HR 1.234; 95% CI 1.184–1.285; *p* < 0.0001). Regardless of which grouping, it was thought that fistulas should be created at least 1 month before beginning hemodialysis, to minimize any intervention to maintain or restore blood flow. Patients who have no history of HTN, but who have a history of DM, CHF, and of using aspirin, clopidogrel and warfarin, were significantly more likely to have fistulas with poorer functional primary patency.

### 3.4. Individualized Functional Maturation Time of Shunts

Forest plots in Figure 5 show hazard ratios in patients with different demographic characteristics. All subgroups demonstrated that fistulas with functional maturation time more than 90 days had superior functional cumulative survivals. This section may be divided by subheadings. It should provide a concise and precise description of the experimental results, their interpretation as well as the experimental conclusions that can be drawn.

## 4. Discussion

In this nationwide population-based, retrospective cohort study, we analyzed 26885 hemodialysis patients with fistulas in Taiwan during a 5-year follow-up period. Functional maturation time of shunts was highly associated with their survival. Fistulas with functional maturation times between 91 and 180 days had the greatest functional cumulative survival. Otherwise, although current popular KDOQI guidance advise that a fistula should be placed at least 6 months before hemodialysis is begun, fistulas with waiting times longer than 180 days for cannulation showed no additional benefit on their survivals in our study [7]. Shunts should be constructed and cannulated at suitable timing, to minimize bridging catheters accompanying complications and for better long-term survival.

According to our results, the ideal functional maturation time for fistulas was between 91 and 180 days on the basis of their best functional cumulative survival. Corpataux et al. [14] demonstrated hemodynamic changes during fistula maturation by using duplex ultrasound to evaluate 6 patients with fistulas in the lower arm. Venous luminal pressure increased initially after shunt creation and gradually returned to normal at 12 weeks, as did venous luminal dilation. Similarly, Wong et al. [15] observed that venous luminal diameter enlarged and reached the plateau at 12 weeks after creation in forearm fistulas, and ultimately the fistulas were successful for hemodialysis. Some believe that it takes about 2–3 months for a vein to complete “arterialization” when an artery is anastomosed to a vein after a fistula creation [16]. In our study, we confirmed the theoretical hemodynamic maturity of fistulas by evaluating clinical data, which showed fistulas with maturation for longer than 90 days is adequate for cannulation. On the other hand, recent one systemic review and meta-analysis [11] reviewed over 8000 studies and included 318 studies comprising 62712 accesses. The mean maturation time for fistulas was 3.5 months in the included studies, but this analysis was limited by the definitions of maturation and patency reported. In our analysis, the functional maturation period of fistulas was certain as at least 90 days, and waiting longer than 180 days had no additional advantage on shunt survival. The mechanical and infectious complications of using bridging catheters could be minimized while fistulas cannulation occurred earlier than 180 days.

Our study showed that the 1-year, 3-year and 5-year functional cumulative survival of fistulas is 81.68%, 74.07%, and 69.54%, respectively. These survival rates are better than previous study results [11,17,18]. One systemic review and meta-analysis [17] analyzed fistulas with inclusion criteria by KDOQI guidance [7], which suggests that fistulas be created at least 6 months prior to hemodialysis. That study observed that the 1-year functional secondary patency rate was 81%; this rate was similar to ours. The other recent meta-analysis [11], which included more studies but inconsistent definitions about accesses maturation, reported the secondary patency at one year was 79%. In Japan, Hatakeyama et al. [18] reported that the 3- and 5-year secondary patency rates were 53% to 59% and 44% to 54%, according to different types of 1560 fistulas. The fistulas were constructed at least 2 to 4 weeks before the initial cannulation for hemodialysis, according to current recommendations in Japan [10]. As compared with previous investigations, our better survival rates in Taiwan imply that functional maturation time plays the key role in functional cumulative survival. According to our results, fistulas should be cannulated between 91 and 180 days after construction for the best functional cumulative survivals.

Among earlier assumptions is the belief that creating a shunt in a timely manner before hemodialysis allows adequate time for shunt maturation. However, creating shunts too early is not without problems. Some patients may develop the steal syndrome due to arterial ischemia in the distal limb, or high-output heart failure, both of which may require additional shunt revision [17,19]. One retrospective study [20] observed that early referral for preparing dialysis aggravated the short- and long-term health-related quality of life (HRQOL) and depression of patients with ESRD. Therefore, we realize that creating shunts too early or too late is not better than using ideal timing.

Compared with other reports in a systemic review and meta-analysis [14] and Japanese records [17,21,22], it is worth noting that functional primary patency of shunts in our study was inferior to other results, although we had superior outcomes in functional cumulative survival. This suggests that the first intervention to maintain or restore blood flow of shunts may be earlier in Taiwan than in other areas. In Taiwan, we evaluate functions of shunts according to the suggestions of KDOQI guidance [7]; thus, earlier interventions to maintain or restore blood flow of shunts may be associated with more aggressive and experienced clinicians, and greater medical convenience by NHI. In another point of view, earlier interventions should be performed to support our excellent functional cumulative survival, but differences in medical costs between earlier interventions and reconstruction of a new shunts need to be clarified.

In our investigation of the association between covariates and shunt survivals, elderly patients, females, patients with DM, CHF, and those using warfarin had significantly inferior functional cumulative survival of fistulas. This finding agrees with the results of previous studies [23,24,25,26]. From another perspective, patients with shunts of functional maturation time more than 180 days, are older, mostly females, with more comorbidities and more likely to use warfarin. We could speculate that old age, female gender, the presence of more comorbidities, and using warfarin could be the negative confounders to shunts maturation. Even though several factors are associated with the maturation and survivals of shunts, fistulas with functional maturation time more than 90 days had superior functional cumulative survivals in all subgroups according to our Forest plots in Figure 5.

In our retrospective analysis, we defined the maturation of fistulas according to clinical evaluation and the rule of 6s characteristics [7]. Postoperative ultrasound of routine care may be a quantitative imaging study in assessing likelihood of AVF maturation and seems reasonable to guide clinicians in decision making [27]. Robbin et al. [28] concluded 6-week ultrasound of AVF blood flow, diameter, and depth are moderately predictive of AVF clinical maturation from the large Hemodialysis Fistula Maturation multicenter study. Subsequent research, including this assessment tool, should evaluate shunt maturity more comprehensively. In our study, it is worth noting that NIHRD covers the most medical records and population in Taiwan. The study population is confirmed by catastrophic illness certification, which minimizes mistakes of diagnostic ICD-9-CM codes and conditions of unexpected shunt cannulation due to acute kidney injury, and can represent almost all the hemodialysis patients. We establish shunts creation, any vascular intervention and shunt abandonment by corresponding operative and therapeutic codes, thus the functional maturation time and shunt survival can be evaluated accurately. Therefore, the results in our study are reliable and really a reflection of shunt condition. Otherwise, there are some limitations. First, in NIHRD, we could not analyze difference between subgroup of AVFs, such as radiocephalic or brachiocephalic shunts. Similarly, clinical parameters, such as anatomical location of shunt, blood pressure, physical status, smoking and body mass index, are not investigated. It should be mentioned that malnutrition-inflammation-atherosclerosis (MIA) syndrome is a feature of ESRD and also a major cause of vascular calcification [29,30], which was correlated with shunt failure. Hypoalbuminemia [31] and C-reactive protein (CRP) [32] are associated with AVF thrombosis indirectly and directly. Laboratory data, especially including albumin and CRP, is necessary in further analysis. Second, the information about comorbidity and medications is only collected from the ICD-9-CM codes claiming for reimbursement, which may be misclassified.

## 5. Conclusions

In our retrospective analysis of NHIRD data, functional maturation time of shunts play a critical role on their survivals. The fistulas with functional maturation time between 91 and 180 days have excellent functional cumulative survivals. The fistulas should be cannulated at least 90 days after their creation, and waiting for longer than 180 days to puncture shows no further benefits regarding their survivals. Further prospective studies may be desired to confirm our perceptions, and the effort on increasing shunts creation before initiation of hemodialysis is also necessary censoriously.

## Figures and Tables

**Figure 1 jcm-08-00247-f001:**
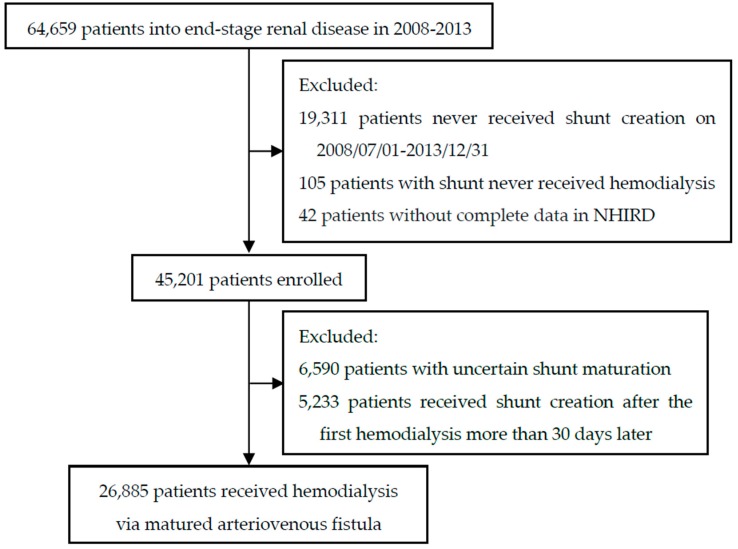
Flow chart of patients enrolled from National Health Insurance Research Database (NHIRD).

**Figure 2 jcm-08-00247-f002:**
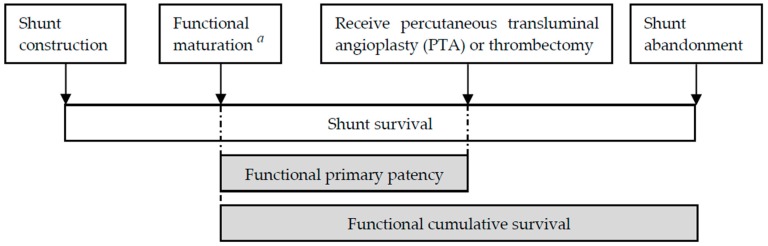
Definitions of shunt survival. *^a^* Functional maturation: shunt is cannulated for hemodialysis successfully.

**Figure 3 jcm-08-00247-f003:**
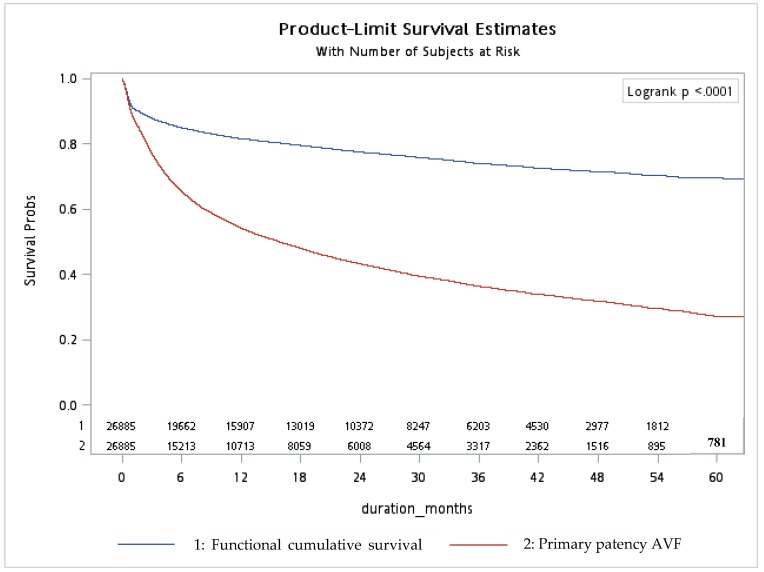
Functional cumulative survival and primary patency of fistulas.

**Figure 4 jcm-08-00247-f004:**
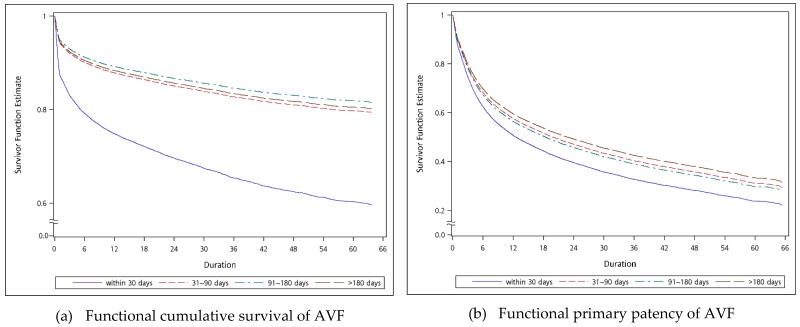
Adjusted Kaplan–Meier plots for (**a**) functional cumulative survival and (**b**) primary patency of fistulas. The model was adjusted for age, gender, history of hypertension, diabetes mellitus, myocardial infarction, congestive heart failure, peripheral vascular disease, cerebrovascular disease, and use of aspirin, clopidogrel, warfarin, and statins.

**Figure 5 jcm-08-00247-f005:**
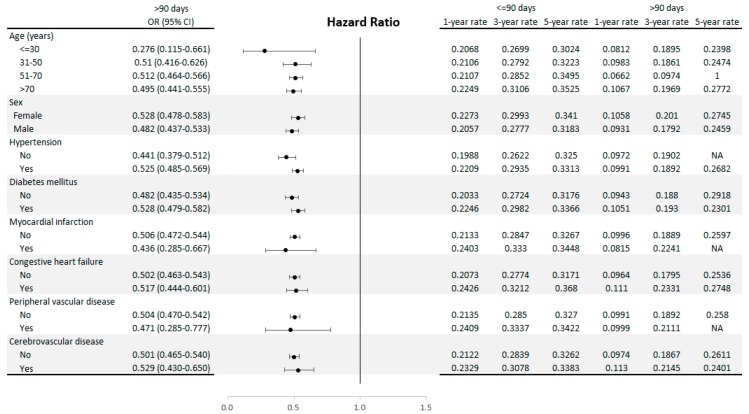
Subgroup hazard ratio for short survival. AVF (arteriovenous fistula) creation [<90 days as reference].

**Table 1 jcm-08-00247-t001:** Characteristics of ESRD patients with AVF (*N* = 26885).

Demographic Characteristics	Functional Maturation Time				*p* Value
	<30 days	31–90 days	91–180 days	>180 days	
Number of populations, *n* (%)	13341 (49.6)	6339 (23.6)	4002 (14.9)	3203 (11.9)	
Men, *n* (%)	8149 (61.1)	3931 (62.0)	2228 (55.7)	1565 (48.9)	<0.0001
Age (year), mean ± SD	62.4 ± 13.3	63.1 ± 13.1	64.0 ± 13.0	66.3 ± 12.5	<0.0001
Age group (year), *n* (%)					<0.0001
<30	188 (1.5)	79 (1.2)	47 (1.2)	20 (0.6)	
31–50	2250 (18.2)	956 (15.1)	527 (13.2)	310 (9.7)	
51–70	6847 (55.4)	3315 (52.3)	2045 (51.1)	1528 (47.7)	
>70	3071 (24.9)	1989 (31.4)	1383 (34.6)	1345 (42.0)	
Comorbidity, *n* (%)					
HTN	9469 (71.0)	4054 (64.0)	2559 (63.9)	2375 (74.2)	<0.0001
DM	6637 (49.8)	2877 (45.4)	1744 (43.6)	1532 (47.8)	<0.0001
MI	438 (3.3)	123 (1.9)	120 (3.0)	71 (2.2)	<0.0001
CHF	2669 (20.0)	1061 (16.7)	675 (16.9)	578 (18.1)	<0.0001
PVD	298 (2.2)	101 (1.6)	50 (1.3)	73 (2.3)	<0.0001
CVD	1212 (9.1)	534 (8.4)	387 (9.7)	349 (10.9)	0.0008
Medication, *n* (%)					
Aspirin	3996 (30.0)	2003 (31.6)	1165 (29.1)	807 (25.2)	<0.0001
Clopidogrel	1570 (11.8)	745 (11.8)	468 (11.7)	405 (12.6)	0.5383
Warfarin	205 (1.5)	108 (1.7)	76 (1.9)	72 (2.2)	0.0332
Statins	4690 (35.2)	2134 (33.7)	1200 (30.0)	765 (23.9)	<0.0001

Abbreviations: ESRD, end stage renal disease; AVF, arteriovenous fistula; SD, standard deviation; HTN, hypertension; DM, diabetes mellitus, MI, myocardial infarction; CHF, congestive heart failure; PVD, peripheral vascular disease; CVD, cerebrovascular disease.

**Table 2 jcm-08-00247-t002:** Hazard ratios for functional cumulative survival of AVF.

	Event, *n* (%)	Crude HR (95% CI)	*p* Value	Adjusted HR (95% CI)	*p* Value
Functional maturation time					
<30 days	4017(30.1)	2.252 (2.097–2.418)	<0.0001	2.245 (2.090–2.411)	<0.0001
31–90 days	934 (14.7)	1.000		1.000	
91–180 days	510 (12.7)	0.895 (0.804–0.997)	0.0445	0.883 (0.792–0.984)	0.0237
>180 days	430 (13.4)	0.988 (0.882–1.108)	0.8400	0.957 (0.853–1.073)	0.4515
Age	5891 (21.9)	1.001 (0.999–1.003)	0.1810	1.002 (1.000–1.004)	0.0275
Gender					
Female	2507 (22.8)	1.000		1.000	
Male	3384 (21.3)	0.942 (0.894–0.992)	0.0226	0.910 (0.864–0.960)	0.0005
HTN					
No	1413 (16.8)	1.000		1.000	
Yes	4478 (24.3)	1.101 (1.037–1.170)	0.0018	0.976 (0.911–1.045)	0.4850
DM					
No	2737 (19.4)	1.000		1.000	
Yes	3154 (24.7)	1.105 (1.050–1.164)	0.0001	1.094 (1.030–1.162)	0.0032
MI					
No	5703 (21.8)	1.000		1.000	
Yes	188 (25.0)	1.148 (0.993–1.327)	0.0627	1.063 (0.912–1.239)	0.4346
CHF					
No	4586 (20.9)	1.000		1.000	
Yes	1305 (26.2)	1.211 (1.139–1.288)	<0.0001	1.150 (1.078–1.228)	<0.0001
PVD					
No	5757 (21.8)	1.000		1.000	
Yes	134 (25.7)	1.143 (0.963–1.356)	0.1272	1.054 (0.887–1.252)	0.5477
CVD					
No	5287 (21.7)	1.000		1.000	
Yes	604 (24.3)	1.082 (0.994–1.177)	0.0676	1.077 (0.987–1.174)	0.0959
Aspirin					
No	4136 (21.9)	1.000		1.000	
Yes	1755(22.0)	1.013 (0.958–1.071)	0.6501	1.000 (0.856–1.019)	0.9978
Clopidogrel					
No	5234 (22.1)	1.000		1.000	
Yes	657 (20.6)	0.965 (0.890–1.047)	0.3922	0.934 (0.856–1.019)	0.1224
Warfarin					
No	5771 (21.8)	1.000		1.000	
Yes	120 (26.0)	1.258 (1.050–1.508)	0.0127	1.285 (1.072–1.541)	0.0067
Statins					
No	4017 (22.2)	1.000		1.000	
Yes	1874 (21.3)	0.943 (0.893–0.997)	0.0370	0.888 (0.838–0.941)	<0.0001

The model was adjusted for age, gender, hypertension, diabetes mellitus, myocardial infarction, congestive heart failure, peripheral vascular disease, cerebrovascular disease, and use of aspirin, clopidogrel, warfarin, and statins. Abbreviations: AVF, arteriovenous fistula; HR, hazard ratios; CI, confidence interval; HTN, hypertension; DM, diabetes mellitus, MI, myocardial infarction; CHF, congestive heart failure; PVD, peripheral vascular disease; CVD, cerebrovascular disease.

**Table 3 jcm-08-00247-t003:** Hazard ratios for functional primary patency of AVF.

	Event, *n* (%)	Crude HR (95% CI)	*p* Value	Adjusted HR (95% CI)	*p* Value
Functional maturation time					
<30 days	7804 (58.5)	1.233 (1.183–1.285)	<0.0001	1.234 (1.184–1.285)	<0.0001
31–90 days	3202 (50.5)	1.000		1.000	
91–180 days	1974 (49.3)	1.040 (0.984–1.100)	0.1658	1.038 (0.981–1.098)	0.1943
180 days	1430 (44.7)	0.947 (0.890–1.008)	0.0892	0.941 (0.884–1.002)	0.0583
Age	14410(53.6)	1.005 (1.004–1.006)	<0.0001	1.005 (1.004–1.006)	<0.0001
Gender					
Female	5938 (53.9)	1.000		1.000	
Male	8472 (53.4)	1.003 (0.970–1.036)	0.8763	0.996 (0.963–1.030)	0.8164
HTN					
No	3563 (24.7)	1.000		1.000	
Yes	10847 (58.8)	1.031 (0.992–1.071)	0.1190	0.941 (0.901–0.983)	0.0065
DM					
No	6663 (47.3)	1.000		1.000	
Yes	7747 (60.6)	1.148 (1.111–1.186)	<0.0001	1.125 (1.082–1.169)	<0.0001
MI					
No	13950 (53.4)	1.000		1.000	
Yes	460 (61.2)	1.182 (1.077–1.297)	0.0004	1.009 (0.915–1.113)	0.8608
CHF					
No	11385 (52.0)	1.000		1.000	
Yes	3025 (60.7)	1.152 (1.107–1.199)	<0.0001	1.081 (1.036–1.128)	0.0003
PVD					
No	14094 (53.5)	1.000		1.000	
Yes	316 (60.5)	1.121 (1.003–1.253)	0.0450	1.045 (0.934–1.169)	0.4381
CVD					
No	12964 (53.1)	1.000		1.000	
Yes	1446 (58.3)	1.070 (1.014–1.130)	0.0144	1.006 (0.952–1.064)	0.8281
Aspirin					
No	9870 (52.2)	1.000		1.000	
Yes	4540 (57.0)	1.157 (1.117–1.199)	<0.0001	1.097 (1.057–1.139)	<0.0001
Clopidogrel					
No	12609 (53.2)	1.000		1.000	
Yes	1801 (56.5)	1.157 (1.101–1.216)	<0.0001	1.061 (1.006–1.119)	0.0302
Warfarin					
No	14130 (53.5)	1.000		1.000	
Yes	280 (60.7)	1.281 (1.138–1.442)	<0.0001	1.248 (1.109–1.406)	0.0002
Statins					
No	9536 (52.7)	1.000		1.000	
Yes	4874 (55.5)	1.055 (1.020–1.092)	0.0022	0.987 (0.952–1.024)	0.4988

The model was adjusted for age, gender, hypertension, diabetes mellitus, myocardial infarction, congestive heart failure, peripheral vascular disease, cerebrovascular disease, aspirin, clopidogrel, warfarin and statins. Abbreviations: AVF, arteriovenous fistula; HR, hazard ratios; CI, confidence interval; HTN, hypertension; DM, diabetes mellitus, MI, myocardial infarction; CHF, congestive heart failure; PVD, peripheral vascular disease; CVD, cerebrovascular disease.

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
