# Peer review of "The Prognostic Significance of Puncture Timing to Survival of Arteriovenous Fistulas in Hemodialysis Patients: A Multicenter Retrospective Cohort Study"

_jcm, 2019, doi:10.3390/jcm8020247_

Reviewer 1 Report

The revised version of Su-Ju Lin's work has further enhanced the already very good work. The graphics and tables are clearly arranged, the information content can be determined quickly and impressively. The discussion is convincing too. The work, in its present form, is valuable in the preemptive placement of shunts in patients with the need for hemodialysis.

Reviewer 2 Report

Authors' revision is enough

This manuscript is a resubmission of an earlier submission. The following is a list of the peer review reports and author responses from that submission.

Round  1

Reviewer 1 Report

The authors provided the association between prognosis and waiting time to the first puncture of AV fistula. I think their data is valuable, but there are a several problems in their analysis.

1.    It is very difficult to understand why the authors divide the object to 4 groups according to days.<30 days group may include many patients without good compliance to medical supply including doctors’ advice. I suggest that 4 groups would be use quartile and calculate mean days in each group.

2.    Malnutrition may effect on fistula failure. Please add serum albumin and CRP in factor analysis.

Reviewer 2 Report

Based on the retrospective analysis of the enormous number of approximately 27,000 hemodialysis patients,  Su-Ju Lin et al conclude that the ideal maturation period for a shunt should be between 90-180 days. This is conclusive from the presented data.

In my opinion, this message is somewhat overstrained in terms of language. It would not hurt the text if that redundancy were tightened a bit.

A weakness of the analysis lies in the methodology: for as much as a retrospective analysis of a collective is suitable for revealing tendencies, other necessities will determine the individual care of a patient. Accordingly, it should be added that there are sensible parameters that indicate an early (re) maturation of the shunt. In particular, postoperative ultrasound measurements are a useful tool that should be integrated into the text as a supplement [1, 2].

1.            Robbin, M.L., et al., Prediction of Arteriovenous Fistula Clinical Maturation from Postoperative Ultrasound Measurements: Findings from the Hemodialysis Fistula Maturation Study. Journal of the American Society of Nephrology, 2018. 29(11): p. 2735-2744.

2.            Oliver, M.J., The Science of Fistula Maturation. Journal of the American Society of Nephrology, 2018. 29(11): p. 2607-2609.